# Mass Transport of Gases across the Air–Water Interface: Implications for Aldehyde Emissions in the Uinta Basin, Utah, USA

**Marc L. Mansfield**

Department of Chemistry and Biochemistry, Utah State University Uintah Basin, 320 North Aggie Boulevard, Vernal, UT 84078, USA; marc.mansfield@usu.edu

**Abstract:** When they dissolve in water, aldehydes become hydrated to gem-diols: $R-COH + H_2O \leftrightarrow RCH(OH)_2$. Such reactions can complicate air–water transport models. Because of a persistent belief that the gem-diols do not exist in the vapor phase, typical models do not allow them to pass through the air–water interface, but in fact, they do. Therefore, transport models that allow both molecular forms to exist in both phases and to pass through the interface are needed. Such a model is presented here as a generalization of Whitman's two-film model. Since Whitman's model has fallen into disuse, justification of its use is also given. There are hypothetical instances for which the flux predicted by the current model is significantly larger than the flux predicted when models forbid the diol form from passing through the interface. However, for formaldehyde and acetaldehyde, the difference is about 6% and 2%, respectively.

**Keywords:** mass transport theory; aldehyde emissions; air–water transport

## 1. Introduction

Exceptionally high ozone concentrations occur during many winters in the Uinta basin of eastern Utah [1–5]. The physics and chemistry of winter ozone formation given the prevailing emissions and meteorology are not completely understood. For example, models routinely underestimate ozone concentrations. Some models suggest that there is a missing source of formaldehyde and other carbonyls [6–8]. This has led us to consider other possible carbonyl sources [9], including produced water evaporation ponds [10–12] and the snowpack [13–15], and to an examination of models of the transport of aldehydes across the air–water interface. Ice crystals in the environment have a surface liquid-like layer. To an initial approximation, transport between snow and the atmosphere resembles transport across the air–liquid water interface [16–18]. This paper is a result of that study.

The theory of mass transport of volatile gases between a liquid and a vapor phase is well developed but becomes more complex when the gas undergoes a reaction in the aqueous phase. Examples include the following reactions:

$$formaldehyde \leftrightarrow methanediol \quad HCHO + H_2O \leftrightarrow H_2C(OH)_2$$

$$acetaldehyde \leftrightarrow 1,1\,ethanediol \quad CH_3COH + H_2O \leftrightarrow CH_3CH(OH)_2$$

$$carbon\ dioxide \leftrightarrow carbonic\ acid \leftrightarrow bicarbonate \leftrightarrow carbonate$$

$$CO_2 + H_2O \leftrightarrow HO(CO)OH \leftrightarrow HCO_3^- + H^+ \leftrightarrow CO_3^{2-} + 2H^+$$

$$dissociation\ of\ HCl\ (and\ other\ acids)\ HCl \leftrightarrow H^+ + Cl^-$$

I will represent all these reactions as 1 ↔ 2, where 1 represents the compound on the left that dominates in the vapor phase. Based on the assumption that the diols, carbonic acid, bicarbonate, carbonate, and chloride do not exist in the vapor phase, it has always been assumed that when form 1 reacts in the aqueous phase to give form 2, then 2 must convert back to 1 while still in the aqueous phase before it can cross into the vapor phase. The primary purpose of this paper is to examine models that challenge this assumption. Of course, the ions do not exist in the vapor phase, but the diols do. Many measurements have detected methanediol vapor in equilibrium with aqueous formaldehyde solutions [19,20]. Methanediol hydrolysis is catalyzed by water and other small molecules, and methanediol has extreme stability in isolation [21–24]. 1,1-Ethanediol in the gas phase has not been studied experimentally, but its reactions are probably similar to methanediol. Once we accept the gas-phase existence of these molecules, we also have to assume that they cross the air–water interface in the diol form. In this paper, I generalize existing models to the case in which both compounds are assumed to exist in both phases and are also assumed to cross the interface. I apply these models to formaldehyde and acetaldehyde transport across the air–water interface.

Interestingly, the existence of carbonic acid molecules in the gas phase has also been established, and they are also predicted to be highly stable in isolation [25–32]; thus, it is also safe to assume that they cross the air–water interface. However, at the acidities expected in the environment, only a small fraction of dissolved carbon dioxide is found in the carbonic acid form; much more is present as free carbon dioxide, bicarbonate, or carbonate. Therefore, when we write 1 ↔ 2 for the carbon dioxide system, we can assume that form 1 is free carbon dioxide, while form 2 is bicarbonate plus carbonate, and we can ignore carbonic acid altogether. Whenever form 2 is ionic, it is appropriate to assume that it does not cross the interface and that existing models suffice. New models are needed for the aldehydes, but not for carbon dioxide or hydrogen chloride.

Mass-transfer theory of transport across the air–water interface has been under development for nearly a century, beginning with Whitman [33] and his "two-film" model. Later decades saw the development of "surface renewal" and "penetration" models [34–39] and arguments based on the Schmidt number [40–42]. The Whitman model is relatively simple and today is used mainly only for pedagogy [43]. However, it was the first model to postulate that to cross the interface, solute molecules must traverse two separate, stagnant films, one on each side of the interface, under conditions in which transport only occurs by molecular diffusivity. It predicts mass-transfer coefficients that scale with molecular diffusivity $D$ to the first power. Later models are more in line with experiments with $D^{1/2}$ or $D^{2/3}$ scaling laws [44–52]. There have also been generalizations that allow both for irreversible [37,53] and reversible [54,55] aqueous phase reactions. However, to the best of my knowledge, no one has ever studied models that allow for reactions in both phases and for both forms to cross the interface.

## 2. Description of the Models

In this paper, I generalize the Whitman model [33] to the case of two reacting species. Five separate models, represented in Figures 1 and 2, will be discussed in this paper. Whitman's model [33] is denoted A1. It models a single compound able to move between the water and air phases. Model A2 was first introduced by Hoover and Berkshire [54]. It permits the reversible reaction 1 ↔ 2 only in the water phase, and only form 1 is able to move across the barrier and exist in the vapor phase. Model A3 assumes that 1 and 2 react reversibly in both phases, but still, only form 1 is able to cross the barrier. Model A4 allows for the reaction in both phases, and it allows for both compounds to cross the barrier. To the best of my knowledge, models A3 and A4 have never been presented in the literature. A fifth model is considered when, without ignoring the existence of both forms of the compound, our analytical techniques do not permit us to distinguish them, and we treat 1 + 2 as a blended compound. Let A1E refer to the model designed to be a single-compound effective representation of A4. Below, I give best techniques for designing an appropriate A1E model.

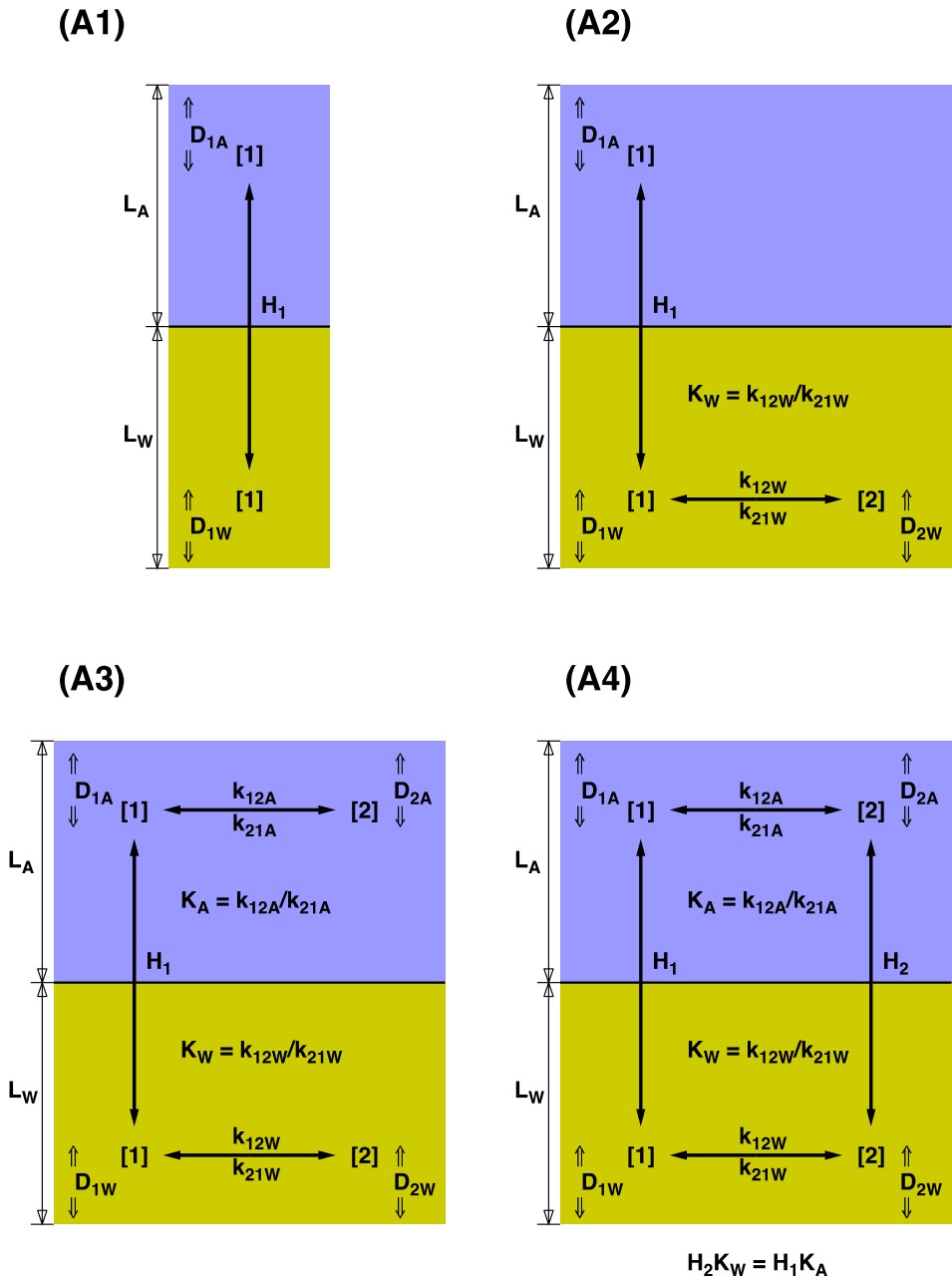

**Figure 1.** Four different models can be defined. A1 treats a single compound that does not react. Models A2, A3, and A4 assume that compound 1 reacts to form compound 2, either only in the water phase or in both phases. Model A4 also assumes that both forms, 1 and 2, are able to cross the interface. A1 is equivalent to Whitman [33], and A2 is equivalent to Hoover and Berkshire [54].

The steady-state reaction–diffusion equations can be solved exactly for all models A1 through A4 if we restrict ourselves to generalizations of the Whitman model [33]. Numerical treatments of model A2 in the context of surface renewal and other more modern models have been considered by Glasscock and Rochelle [55]. Despite yielding exact solutions, each succeeding model is algebraically and numerically more complex than the previous. Therefore, the exact solutions will only be summarized here; details and derivations are given in the Supplementary Material.

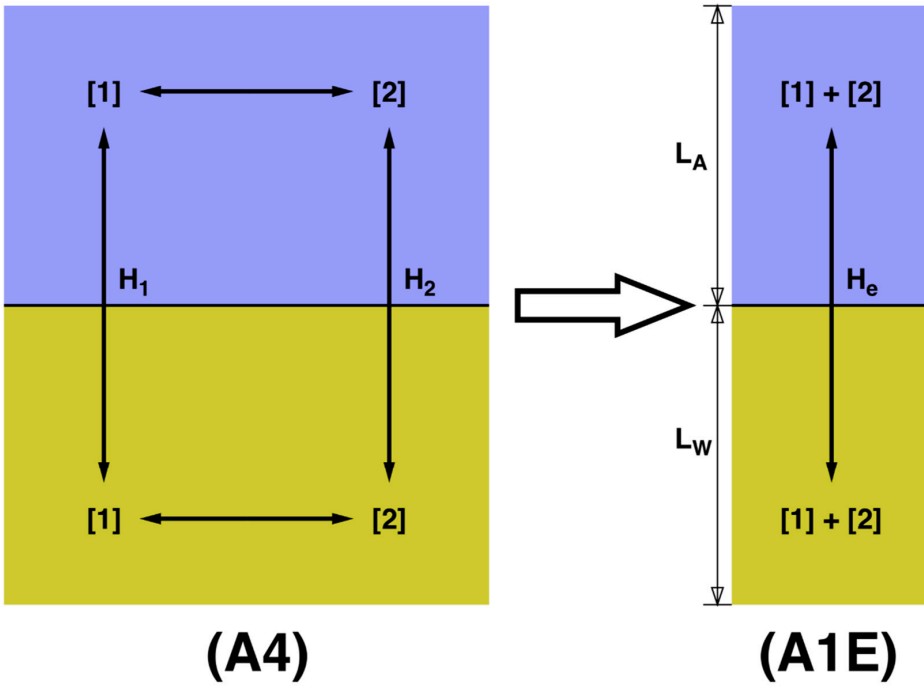

**Figure 2.** A fifth model, A1E, can be considered if we treat 1 + 2 as a blended compound.

## 3. Definition of Variables

Table 1 defines the variables employed. We distinguish between "system" variables that define the physical and chemical properties of the system, "concentration" variables that represent concentrations of the two compounds at specific depths, and "derived" variables, used for notational convenience. The subscripts "1" and "2" identify variables that are specific to compounds 1 and 2, respectively. The subscripts "A" and "W" identify variables specific to the air and water phases, respectively. The subscript "t" for total represents a sum of properties over both compounds, e.g., $F_t = F_1 + F_2$ is the total flux. The subscripts "0" and "$\infty$" attached to concentration variables indicate interfacial and far-field concentrations, respectively. A generic expression valid for either phase or for either compound can appear in this paper without the relevant subscripts.

The reaction rate constants $k_{12}$ and $k_{21}$ are pseudo-first order, since variations in water concentration are negligible. $k_{12}$ refers to the rate of $1 \rightarrow 2$ and vice versa. Enhancement factors appear in models A2 and A3 as multipliers of certain transfer coefficients. As defined, enhancement factors are always greater than 1, so these models predict that reactions can accelerate transfer. Because diffusivities of small molecules agree better than an order of magnitude, we can usually expect that $Q \approx 1$. Because compounds 1 and 2 usually do not have the same mass, molar concentrations are used throughout. The equilibrium constants are defined as $K = $ [diol]/[aldehyde]. I use the dimensionless air-over-water form of Henry's constant.

$$H_1 = \frac{C_{1A,eq}}{C_{1W,eq}} \, , \, H_2 = \frac{C_{2A,eq}}{C_{2W,eq}} \tag{1}$$

with the subscript "eq" denoting equilibrium concentrations. It is important to distinguish between "intrinsic" and "effective" Henry's constants. $H_1$ and $H_2$ as they are used here are intrinsic. $H_1$ is the ratio of aldehyde concentrations only, excluding the gem-diol form. $H_2$ is the ratio of only the gem-diol concentrations. Effective Henry's constants blur the distinction between the aldehyde and the gem-diol forms:

$$H_e = \frac{C_{1\infty A} + C_{2\infty A}}{C_{1\infty W} + C_{2\infty W}} = \frac{H_1 + H_2 K_W}{1 + K_W} = \frac{H_1(1 + K_A)}{(1 + K_W)} \tag{2}$$

Effective Henry's constants appear commonly in the literature, especially when measurement techniques are unable to distinguish the two forms of the molecule. Not all the variables given in Table 1 are independent. Interrelations between variables are listed in Table 2. These interrelations are results of physical laws or of fundamental assumptions in the models.

**Table 1.** List of system and concentration variables.

| Notation | Description | Units |
|---|---|---|
| | SYSTEM VARIABLES | |
| $D_{1A}$, $D_{2A}$, $D_{1W}$, $D_{2W}$ | Molecular diffusivities | $cm^2\ s^{-1}$ |
| $k_{12A}$, $k_{21A}$, $k_{12W}$, $k_{21W}$ | Reaction rate constants | $s^{-1}$ |
| $K_A$, $K_W$ | Equilibrium constants | dimensionless |
| $H_1$, $H_2$ | Intrinsic Henry's constants | dimensionless |
| $H_e$ | Effective Henry's constant | dimensionless |
| $L_A$, $L_W$ | Film thicknesses | cm |
| $F_1$, $F_2$, $F_t$ | Fluxes through the interface | $mol\ cm^{-2}\ s^{-1}$ |
| $f_1$, $f_2$, $f_t$ | Mass transfer coefficients | $cm\ s^{-1}$ |
| | CONCENTRATION VARIABLES | |
| $C_{1\infty W}$, $C_{1\infty A}$, $C_{2\infty W}$, $C_{2\infty A}$ $C_{10W}$, $C_{10A}$, $C_{20W}$, $C_{20A}$ | Concentrations at specific depths | $mol\ cm^{-3}$ |
| | DERIVED VARIABLES | |
| $Q = \frac{D_1}{D_2}$ | Diffusivity ratios | dimensionless |
| $m = \frac{C_{1\infty A}}{H_1 C_{1\infty W}}$ | Equilibrium indicator | dimensionless |
| $d_1 = \left(\frac{D_1}{k_{12}}\right)^{1/2}$ $d_2 = \left(\frac{D_2}{k_{21}}\right)^{1/2}$ $d = \left(d_1^{-2} + d_2^{-2}\right)^{-1/2}$ | Reaction–diffusion distances | cm |
| $\Lambda = \frac{L}{d}$ | Reduced film thickness | dimensionless |
| $\zeta_A = \frac{D_{2A}}{L_A}$, $\zeta_W = \frac{D_{2W}}{L_W}$ $Q_A \zeta_A = \frac{D_{1A}}{L_A}$, $Q_W \zeta_W = \frac{D_{1W}}{L_W}$ | Zeta-notation | $cm\ s^{-1}$ |
| $E = \frac{\Lambda(Q+K)}{\Lambda Q + K \tanh\Lambda}$ | Enhancement factors | dimensionless |

**Table 2.** Relationships between variables reduce the number of degrees of freedom in any model.

| RELATIONSHIP | JUSTIFICATION |
|---|---|
| $K_A = \frac{k_{12A}}{k_{21A}}$ $K_W = \frac{k_{12W}}{k_{21W}}$ | Equilibrium constants are ratios of forward and reverse rate constants. |
| $H_2 K_W = H_1 K_A$ | Required by detailed balance. |
| $C_{2\infty W} = K_W\, C_{1\infty W}$ $C_{2\infty A} = K_A\, C_{1\infty A}$ | Assumption of far-field chemical equilibrium. |
| $C_{10A} = H_1\, C_{10W}$ $C_{20A} = H_2\, C_{20W}$ | Assumption of rapid exchange through the interface and local equilibrium across the interface. |

## 4. Justification of the Whitman Model

Because the Whitman model [33] has fallen into disuse, I now justify using generalizations of it in this study. The essential argument is that when Whitman's film thickness is chosen to duplicate the flux of any of the more recent models, we can expect that it gives the same qualitative dynamics, the difference in *D*-scaling notwithstanding. Fundamentally, the difference between the Whitman model and, for example, the Schmidt number models is the assumed mathematical form of the eddy diffusivity. For Whitman, we have

$$E(z) = \begin{cases} D, \; if \; 0 < z < L \\ \infty, \; if \; z > L \end{cases} \tag{3}$$

The eddy diffusivity for the Schmidt number models can be written as

$$E(z) = D\left[1 + \left(\frac{z}{L'}\right)^p\right] \tag{4}$$

$L'$ is a lumped quantity with units of length. Physical arguments require $p = 2$ or $3$ [40–42]. Interestingly, Equation (3) is the large-*p* version of Equation (4). Calculation of the transfer coefficient involves the integral

$$k^{-1} = \int_0^\infty \frac{dz}{E(z)} \tag{5}$$

The integrands $1/E(z)$ of Equations (3) and (4) are both plotted in Figure 3. $L'$ and $1/D$ are measures, respectively, of the breadth and height of the area under the blue curve, meaning that Equation (4) gives this scaling law: $k \sim D/L'$. Equation (3) implies $k = D/L$. Therefore, when properly chosen, there is a value of $L \cong L'$ for which both models give the same $k$. Because the Whitman model has the same effective film thickness as the Schmidt number model, and because its diffusivity is approximately equal to the eddy diffusivity in the film, we expect that the two models give the same qualitative dynamics.

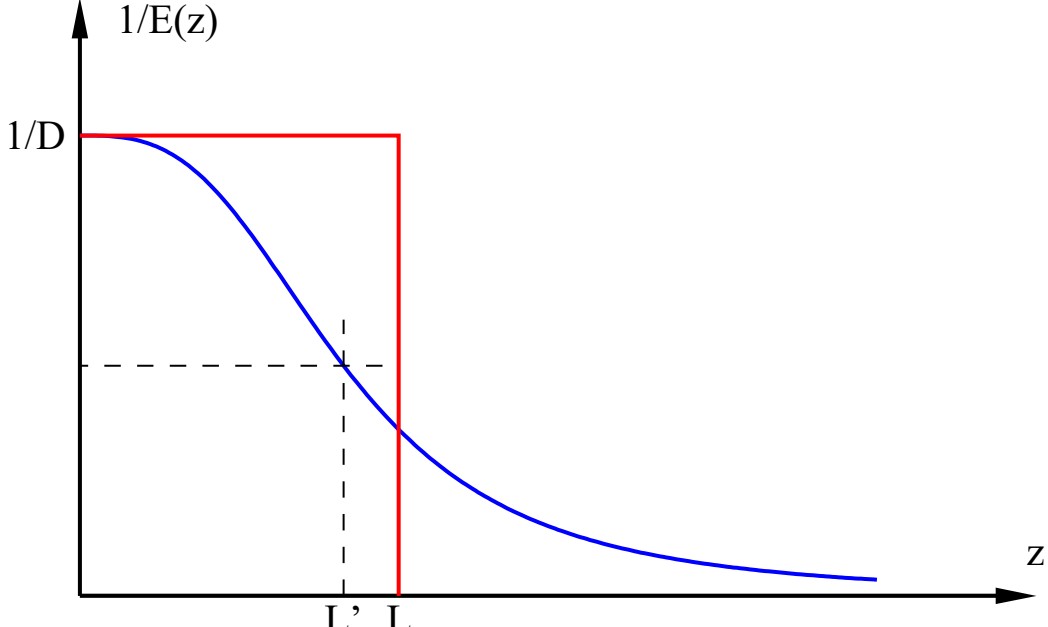

**Figure 3.** The red rectangle and blue curve are $1/E(z)$ for $E(z)$ as given in Equations (3) and (4), respectively. They are drawn with the same area under the curves. $L'$ is the width at half-height of the blue curve; the area under the blue curve scales as $L'/D$. The red rectangle has area $L/D$.

However, the scaling law $k \sim D/L'$ does not seem to be appropriate for the Schmidt number models, which give $k \sim D^{1/2}$ or $D^{2/3}$. The explanation of the discrepancy is that $L'$ is indeed a measure of film thickness, but as a lumped variable, it has its own $D$ dependence. The scaling law for the Schmidt number models emerges from the assumptions (1) that $E(0) = D$, and (2) that the term in $z^p$ is determined only by fluid-dynamic properties of the solvent; no property of the solute, including $D$, is assumed to enter. Rather, the coefficient of the term in $z^p$ is assumed to depend only on three parameters, solvent viscosity $\eta$, solvent density $\rho$, and friction velocity at the interface, $u^*$ [42]. For any given $p$, there is only one combination of these three parameters that yields the correct units. The resultant scaling laws for $L'$ and $k$ are:

$$L' \sim \left(\frac{\eta}{\rho}\right)^{(p-1)/p} (u^*)^{-1} D^{1/p} \tag{6}$$

$$k \sim (Sc)^{-n} u^* \text{ with } n = (p-1)/p \tag{7}$$

Here, $Sc = \eta/\rho D$ is the Schmidt number of the solvent; $u^*$ is the friction velocity; and $n$ is 1/2 and 2/3 respectively, when $p = 2$ or 3. Dependence on wind speed enters $L'$ through the parameter $u^*$.

## 5. Properties of the Models

See the Supplementary Material for derivations of all the models and a detailed discussion of their properties. In this section, I summarize some of the more important properties of the models.

### 5.1. Independent Concentration Variables and Equilibrium Conditions

Because of the interrelations between concentration variables listed in Table 2, only two concentration variables are independent in any of the models. For model A1, $C_{\infty W}$ and $C_{\infty A}$ are the logical choices for independent concentration variables, while for the other models, $C_{1\infty W}$ and $C_{1\infty A}$ make the most sense. This is because the most natural boundary conditions on the problem are the specification of the far-field concentrations. Then $C_{2\infty W}$ and $C_{2\infty A}$ are fixed by the far-field equilibrium assumption, Table 2, Row 3. The ratios $C_{10A}/C_{10W}$ and $C_{20A}/C_{20W}$ are fixed by the assumption of rapid equilibration across the interface, Table 2, Row 4. The values of $C_{10W}$ and $C_{20W}$ are fixed by the steady-state requirement that all fluxes through the interface be balanced.

Furthermore, I define the following dimensionless ratio

$$m = \frac{C_{1\infty A}}{H_1 C_{1\infty W}} \tag{8}$$

or, for model A1,

$$m = \frac{C_{\infty A}}{H C_{\infty W}} \tag{9}$$

I use $m$ to replace dependence on $C_{1\infty A}$. $m$ is useful because it indicates whether air or water concentrations are in excess and it indicates the direction of flow between the two phases. For this reason, I call it the "equilibrium indicator." In any of the models, the total flux through the interface can be factored into three terms:

$$F = (1-m)C_{1\infty W} f \tag{10}$$

I observe the convention that positive flux flows from the water to the air. Then $m < 1$ corresponds to net positive flux, $m > 1$ to net negative flux, and $m = 1$ corresponds to equilibrium with zero flux. $f$ plays the role of a mass-transfer coefficient and has units of velocity. It will be written with subscripts "A1," "A2," etc., to distinguish between the different models. However, other definitions of the mass-transfer coefficients are found in the literature; for examples, see Section 7.3.

### 5.2. Converting Model A1 to A1E

Here I give the complete description of model A1E. The total flux for the original Whitman model, model A1 (see the Supplementary Material), is

$$F = \frac{k_W k_A}{(H k_A + k_W)}(H C_{\infty W} - C_{\infty A})$$

with

$$k_W = \frac{D_W}{L_W}, \quad k_A = \frac{D_A}{L_A}, \quad H = \frac{C_{A0}}{C_{W0}}$$

To obtain model A1E, use the effective Henry's constant, Equation (2), and weighted averages of the diffusivities:

$$D_{eA} = \frac{D_{1A} + K_A D_{2A}}{1 + K_A}$$

$$D_{eW} = \frac{D_{1W} + K_W D_{2W}}{1 + K_W}$$

$$\zeta_{eA} = \frac{D_{eA}}{L_A} = \zeta_A \left( \frac{Q_A + K_A}{1 + K_A} \right)$$

$$\zeta_{eW} = \frac{D_{eW}}{L_W} = \zeta_W \left( \frac{Q_W + K_W}{1 + K_W} \right)$$

and write this for the flux:

$$F_{A1E} = \frac{\zeta_{eW} \zeta_{eA}}{(H_e \zeta_{eA} + \zeta_{eW})}[H_e(C_{1\infty W} + C_{2\infty W}) - (C_{1\infty A} + C_{2\infty A})]$$

With these substitutions: $C_{1\infty A} = m H_1 C_{1\infty W}$, $C_{2\infty W} = K_W C_{1\infty W}$, $C_{2\infty A} = K_A C_{1\infty A}$, we can write

$$F_{A1E} = f_{A1E}(1 - m) C_{1\infty W}$$

where

$$f_{A1E} = \frac{\zeta_{eW} \zeta_{eA}}{(H_e \zeta_{eA} + \zeta_{eW})} H_1 (1 + K_A)$$

or

$$\frac{1}{f_{A1E}} = \frac{1}{(Q_A + K_A) H_1 \zeta_A} + \frac{1}{(Q_W + K_W) \zeta_W} \tag{11}$$

### 5.3. Properties of the Models

All the following properties are established fully in the supplementary information.

(1) Whitman showed us [33] that one or the other of the two films is often rate-determining. In such cases, the mass-transfer coefficients depend only on system parameters of the rate-limiting film, and we speak of "air-layer" or "water-layer control". $\zeta_A \ll \zeta_W$ produces air control and vice versa. The two variables $R_A$ and $R_W$ satisfy $R_A + R_W = 1$ (see the supplementary information) and indicate the extent of control by one side or the other. When $R_A \cong 1$, the system is under air-barrier control; when $R_W \cong 1$, it is under water barrier control.

(2) In all cases, $f_{A1} < f_{A2} < f_{A3} < f_{A4} < f_{A1E}$, but there are limiting cases in which any two may become arbitrarily close. The first two inequalities are not surprising; we expect any kinetic process to speed up when new pathways open up. $f_{A1E} \cong f_{A4}$ often occurs when the system is under either air- or water-barrier control. It also occurs when the reactions are so fast that the two molecular forms have no individual identity.

(3) $f_{A3} < f_{A4}$ always, but they often agree well. Good agreement occurs, for example, whenever flux of 2 through the interface is negligible. However, poor agreement is also observed,

indicating that there are conditions in which models A2 or A3 are inadequate. The passage of both forms through the interface should not generally be ignored.

(4) Instances of $f_{A1E} \cong f_{A4}$ or $f_{A3} \cong f_{A4}$ are beneficial because of the numerical complexity of model A4. Unfortunately, it is generally not possible to know when the approximations are valid without actually performing the calculations.

(5) The reaction–diffusion distances $d_1$, $d_2$, and $d$ defined in Table 1 measure the typical distance over which molecules diffuse before they react. The dimensionless ratio $\Lambda$ distinguishes between systems with fast or slow reactions: if $\Lambda \ll 1$, then molecules do not interconvert as they diffuse through the film, while if $\Lambda \gg 1$, they interconvert many times.

(6) Results of all models simplify in certain limits summarized in Table 3. Models A2 and A3 both involve enhancement factors that change from $E \cong 1$ (no enhancement) when $\Lambda \ll 1$, to $E \cong (Q + K)/Q$ when $\Lambda \gg 1$. The enhancement factors can be very large when $K \gg 1$. In model A4, the enhancement terms do not directly factor out of the formulas, but enhancement is occurring as can be seen in Table 3. However, in model A4, the magnitude of the enhancement does not evolve with changing $\Lambda$, but remains near $(Q + K)/Q$ for all $\Lambda$.

**Table 3.** Results of all models simplify in certain limits.

| Limiting Value | Air-Side Control $\zeta_W \gg \zeta_A$ | | Water-Side Control $\zeta_W \ll \zeta_A$ | |
|---|---|---|---|---|
| $f_{A1E} \cong$ | $\zeta_A H_1 (Q_A + K_A)$ | | $\zeta_W (Q_W + K_W)$ | |
| $f_{A2} \cong$ | $\zeta_A H_1 Q_A$ | | $\zeta_W Q_W E_W$ | |
| | | | $(\Lambda_W \ll 1)$ $\zeta_W Q_W$ | $(\Lambda_W \gg 1)$ $\zeta_W (Q_W + K_W)$ |
| $f_{A3} \cong$ | $\zeta_A H_1 Q_A E_A$ | | $\zeta_W Q_W E_W$ | |
| | $(\Lambda_A \ll 1)$ $\zeta_A H_1 Q_A$ | $(\Lambda_A \gg 1)$ $\zeta_A H_1 (Q_A + K_A)$ | $(\Lambda_W \ll 1)$ $\zeta_W Q_W$ | $(\Lambda_W \gg 1)$ $\zeta_W (Q_W + K_W)$ |
| $f_{A4} \cong$ | $\zeta_A H_1 (Q_A + K_A)$ | | $\zeta_W (Q_W + K_W)$ | |

(7) Table 3 lets us identify several regimes for which model A3 and A4 differ significantly; for example, under air-side control when $\Lambda_A \ll 1$ and $K_A \gg 1$, and under water-side control when $\Lambda_W \ll 1$ and $K_W \gg 1$.

## 6. Aldehyde ↔ Gem-Diol Reactions

We consider the hydration reaction of formaldehyde to form methanediol:

$$HCHO + H_2O \leftrightarrow CH_2(OH)_2$$

and of acetaldehyde to form 1,1-ethanediol:

$$CH_3 - COH + H_2O \leftrightarrow CH_3 - CH(OH)_2$$

In aqueous solution, the diols can continue to add aldehyde units to form higher oligomers [56], but at the aldehyde and diol concentrations expected in the environment, those compounds can be neglected [57].

According to theoretical estimates [21,23] the unimolecular decay rate of methanediol is very low, giving a half-life at 300 K in an extreme vacuum longer than the age of the universe. However, in the atmosphere and in the aqueous phase, the reaction is catalyzed by $H_2O$, by organic and inorganic acids, by bases, and by the hydroperoxyl radical. The reaction is probably also autocatalytic [22,58–60]. Calculations also indicate that the acid and radical catalysts may be more efficient than water, although the relative abundance of water still means that it is the most important catalyst.

A half-century ago, Eigen [61] suggested that water catalysis proceeds via a concerted exchange of protons involving two water molecules. That view has been confirmed theoretically, along with the suggestion that three-water catalysis is even faster [21,23,62,63]. These findings imply that the absolute rate constants are probably second or third order in water. The gas-phase pseudo-first-order reaction rates $k_{12A}$ and $k_{21A}$ are then expected to be strong functions of the concentrations of any catalysts and to be slower than the equivalent reactions in the aqueous phase where the catalyst concentration is larger. The air film at the interface is expected to be water saturated, which translates into a strong temperature dependence for $k_{12A}$ and $k_{21A}$ arising from the dependence of absolute humidity on temperature. The aqueous reactions are also catalyzed by $H_3O^+$ and $OH^-$ so that $k_{12W}$ and $k_{21W}$ are pH dependent, and their values are lowest at about pH 7.

The rigorously defined gas-phase equilibrium constant for the reaction aldehyde + $H_2O$ ↔ diol is written in terms of the partial pressures of each gas as

$$\kappa_A = \frac{P_{diol}\, P_0}{P_{aldehyde}\, P_{H_2O}}$$

where $P_0$ is a reference pressure of 1 atm. The expression we use for $K_A$ is

$$K_A = \frac{[diol]}{[aldehyde]} = \frac{P_{diol}}{P_{aldehyde}} = \left(\frac{P_{H_2O}}{P_0}\right)\kappa_A$$

meaning that literature values of $\kappa_A$ must be corrected using the partial pressure of water vapor (0.039 atm at 298.15 K). The rigorous equilibrium constant has strong temperature dependence [64,65], but so does the partial pressure of water vapor [66]. Interestingly, the two effects largely compensate, and $K_A$ for the formaldehyde reaction is essentially independent of temperature, as shown in Figure 4. The effect of relative humidity on $K_A$ is also displayed in Figure 4. Of course, in the air film at the air–water interface, 100% relative humidity can be assumed.

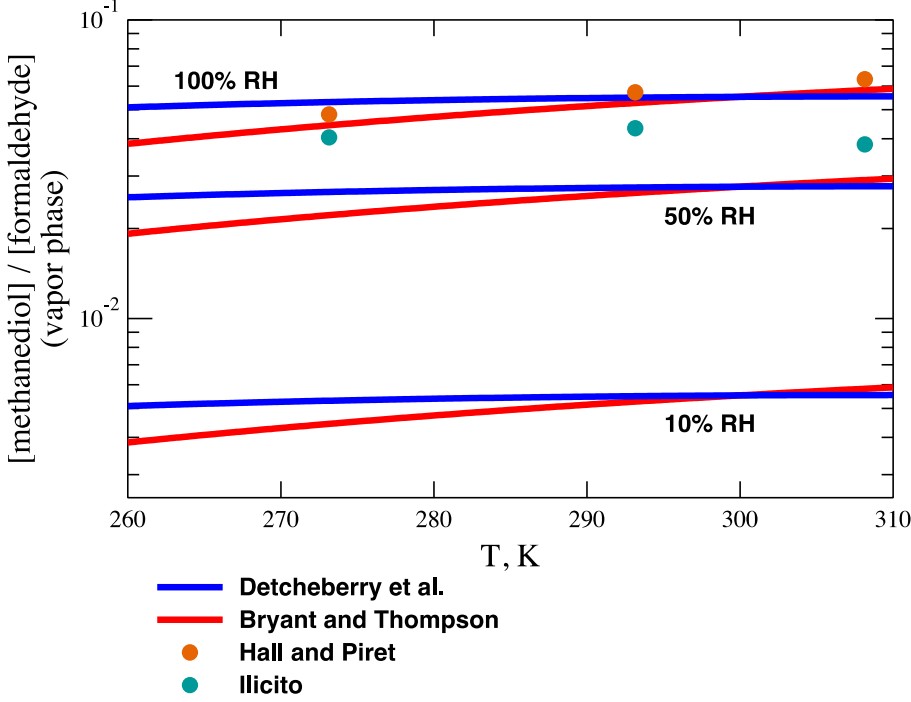

**Figure 4.** Temperature dependence of the formaldehyde $K_A$. Data are from Detcheberry et al., Bryant and Thompson, and from Iliceto and Hall and Piret as quoted by Bryant and Thompson [64,65]. The saturation vapor pressure of water was calculated according to [66]. The red and blue traces were also calculated assuming three different values of relative humidity.

Tables 4 and 5 contain literature data on each aldehyde reaction. Henry's constants $H_1$ for the aldehydes have been retrieved from Sander's compilation [67]. $H_2$ for the gem-diols does not appear in his compilation, but it is constrained by the values of $H_1$, $K_A$, and $K_W$. Henry's constants are often difficult to measure, and there is usually considerable scatter in Sander's compilation for any one compound, often by more than an order of magnitude. Here, I take the median Henry's constants from the compilation, which appear to be good consensus values. Sander tabulates effective Henry's constants, $H_e$, for formaldehyde and acetaldehyde. Therefore, his entries must also be converted to intrinsic constants.

Molecular diffusivities were calculated using several different empirical correlations [68–71]. The quantities $K_W$, $k_{12W}$ and $k_{21W}$ have been extensively measured and calculated theoretically. For these quantities, Tables 4 and 5 report ranges of values equal to the middle half of datasets of independently measured or computed values. The value of the formaldehyde $K_A$ is taken from Figure 4 at 100% relative humidity.

**Table 4.** Data relevant to the $HCHO \leftrightarrow CH_2(OH)_2$ reaction at $\approx 300$ K and pH 7.

| Variable | Value or Range | Source |
|----------|----------------|--------|
| $H_1$ | 0.025 | Median from [67], corrected to intrinsic. |
| $H_2$ | $\approx 10^{-7}$ | Never measured, but constrained by $H_1$, $K_A$, $K_W$. |
| $D_{1W}$ | $1.87 \times 10^{-5}$ cm$^2$ s$^{-1}$ | [68,70,71] |
| $D_{2W}$ | $1.57 \times 10^{-5}$ cm$^2$ s$^{-1}$ | [68,70,71] |
| $D_{1A}$ | 0.155 cm$^2$ s$^{-1}$ | [69,71] |
| $D_{2A}$ | 0.124 cm$^2$ s$^{-1}$ | [69,71] |
| $K_A$ | 0.045 | Figure 4. |
| $K_W$ | 1800 to 2270 | Middle seven of thirteen independent datapoints (experimental and theoretical) [63,64,72–76] |
| $L_A$ | 0.3 cm | [43] |
| $L_W$ | 0.02 cm | [43] |
| $k_{12W}$ | 9.8 s$^{-1}$ to 10.6 s$^{-1}$ | Middle four of six independent datapoints (experimental and theoretical) [74,75,77] |
| $k_{21W}$ | $(4.5$ to $5.3) \times 10^{-3}$ s$^{-1}$ | Middle four of six independent datapoints (experimental and theoretical) [59,75,78] |
| $k_{12A}$ | Unknown | Never measured, but ratio $k_{12A}/k_{21A}$ is constrained. |
| $k_{21A}$ | Unknown | Never measured, but ratio $k_{12A}/k_{21A}$ is constrained. |

**Table 5.** Data relevant to the $CH_3 - COH \leftrightarrow CH_3 - CH(OH)_2$ reaction at $\approx 300$ K and pH 7.

| Variable | Value or Range | Source |
|---|---|---|
| $H_1$ | $7.0 \times 10^{-3}$ | Median from [67] corrected to intrinsic. |
| $H_2$ | $\approx 10^{-5}$ | Never measured, but constrained by $H_1$, $K_A$, $K_W$. |
| $D_{1A}$ | $0.119$ cm$^2$ s$^{-1}$ | [69,71] |
| $D_{2A}$ | $0.103$ cm$^2$ s$^{-1}$ | [69,71] |
| $D_{1W}$ | $1.46 \times 10^{-5}$ cm$^2$ s$^{-1}$ | [68,70,71] |
| $D_{2W}$ | $1.29 \times 10^{-5}$ cm$^2$ s$^{-1}$ | [68,70,71] |
| $K_A$ | $\approx 0.005$ | Poorly constrained. See text. |
| $K_W$ | 1.19 to 1.35 | The middle thirteen of 27 independent datapoints (experimental and theoretical) [73,76,79–82] |
| $L_A$ | 0.3 cm | [43] |
| $L_W$ | 0.02 cm | [43] |
| $k_{12A}$ | Unknown | Never measured, but ratio $k_{12A}/k_{21A}$ is constrained. |
| $k_{21A}$ | Unknown | Never measured, but ratio $k_{12A}/k_{21A}$ is constrained. |
| $k_{12W}$ | $(0.005 \text{ to } 0.015)$ s$^{-1}$ | Middle four of eight independent datapoints (experimental and theoretical) [75,77]. |
| $k_{21W}$ | $(0.0041 \text{ to } 0.012)$ s$^{-1}$ | Middle four of eight independent datapoints (experimental and theoretical) [75,80]. |

Apparently, the only publication considering the equilibrium constant of the vapor-phase hydration of acetaldehyde is a theoretical calculation by Rayne and Forest [83] obtaining $\kappa_A \approx 0.005$ (geometric mean of three separate estimates) at 298.15 K. For formaldehyde at 298.15 K, Rayne and Forest obtain $\kappa_A \approx 0.07$ (geometric mean of three separate estimates), whereas experimental values are around 1.75 [64], i.e., the Rayne–Forest formaldehyde estimate is a factor of about 25 too low. Applying the same ratio to acetaldehyde yields $\kappa_A \approx 0.12$ and $K_A \approx 0.005$. Therefore, in the following, we will estimate $K_A \approx 0.005$ for acetaldehyde, all the while considering this to be a poorly constrained value.

The rates $k_{12A}$ and $k_{21A}$ have never been measured for either system, but their ratio is constrained by $K_A$. Therefore, the only available constraints on $k_{12A}$ and $k_{21A}$ are $k_{21A} \lesssim k_{21W}$ and $k_{12A} \lesssim k_{12W}$, as explained above. In calculations reported below, I have allowed $k_{12A}$ and $k_{21A}$ to vary by about four orders of magnitude while enforcing these constraints.

## 7. Results

### 7.1. Numerical Results for Formaldehyde and Acetaldehyde

Table 6 summarizes numerical calculations for formaldehyde and acetaldehyde assuming the values given in Tables 4 and 5. Because it is poorly constrained, $k_{21A}$ was allowed to vary from $10^{-6}$ s$^{-1}$ to $10^{-2.5}$ s$^{-1}$, but the final results for $f$ are insensitive to this variation. Neither the air- nor the water-side dominates in either system ($R_A = 0.544$ or 0.234). $f_{A1E}$ is 87% and 58% larger than $f_{A4}$ for formaldehyde and acetaldehyde, respectively, so model A1E is not accurate in these cases. The ratios $f_{A2}/f_{A4}$ and $f_{A3}/f_{A4}$ are, respectively, 0.935 and 0.984, so model A3 is adequate for formaldehyde and acetaldehyde.

**Table 6.** Sample calculations. $k_{21A}$ was allowed to assume eight different values as indicated. Each of these generated eight different values for $k_{12A}$, $d_A$, and $\Lambda_A$, but all remaining variables were insensitive to these variations, at least to the number of significant figures given.

| Variable | Formaldehyde | Acetaldehyde |
|---|---|---|
| $H_1$ | 0.025 | $7.0 \times 10^{-3}$ |
| $D_{1W}/(\text{cm}^2 \text{ s}^{-1})$ | $1.87 \times 10^{-5}$ | $1.46 \times 10^{-5}$ |
| $D_{2W}/(\text{cm}^2 \text{ s}^{-1})$ | $1.57 \times 10^{-5}$ | $1.29 \times 10^{-5}$ |
| $D_{1A}/(\text{cm}^2 \text{ s}^{-1})$ | 0.155 | 0.119 |
| $D_{2A}/(\text{cm}^2 \text{ s}^{-1})$ | 0.124 | 0.103 |
| $k_{12W}/\text{s}^{-1}$ | 10.0 | 0.01 |
| $k_{21W}/\text{s}^{-1}$ | $5.0 \times 10^{-3}$ | 0.008 |
| $K_A$ | 0.045 | 0.005 |
| $L_W/\text{cm}$ | 0.02 | 0.02 |
| $L_A/\text{cm}$ | 0.3 | 0.3 |
| $k_{21A}/\text{s}^{-1}$ | $\{10^{-X}\}$[a] | $\{10^{-X}\}$[a] |
| $K_W$ | 2000 | 1.25 |
| $k_{12A}/\text{s}^{-1}$ | $0.045 \times \{10^{-X}\}$[a] | $0.005 \times \{10^{-X}\}$[a] |
| $H_2$ | $5.625 \times 10^{-7}$ | $2.8 \times 10^{-5}$ |
| $d_A/\text{cm}$ | {6.15, 10.9, 19.5, 34.6, 61.5, 109, 195, 346} | {5.69, 10.1, 18.0, 32.0, 56.9, 101, 180, 320} |
| $d_W/\text{cm}$ | $1.37 \times 10^{-3}$ | $2.77 \times 10^{-2}$ |
| $Q_A$ | 1.25 | 1.16 |
| $Q_W$ | 1.19 | 1.13 |
| $\Lambda_A$ | $\{488, 274, 154, 86.7, 48.8, 27.4, 15.4, 8.67\} \times 10^{-4}$ | $\{527, 296, 167, 93.7, 52.7, 29.6, 16.7, 9.37\} \times 10^{-4}$ |
| $\Lambda_W$ | 14.6 | 0.723 |
| $H_e$ | $1.3 \times 10^{-5}$ | $3.1 \times 10^{-3}$ |
| $\zeta_A/(\text{cm s}^{-1})$ | 0.413 | 0.343 |
| $\zeta_W/(\text{cm s}^{-1})$ | $7.85 \times 10^{-4}$ | $6.45 \times 10^{-4}$ |
| $R_A$ | 0.544 | 0.234 |
| $f_{A1E}/(\text{cm s}^{-1})$ | 0.0133 | $9.91 \times 10^{-4}$ |
| $f_{A4}/(\text{cm s}^{-1})$ | $7.08 \times 10^{-3}$ | $6.25 \times 10^{-4}$ |
| $f_{A3}/(\text{cm s}^{-1})$ | $6.62 \times 10^{-3}$ | $6.15 \times 10^{-4}$ |
| $f_{A2}/(\text{cm s}^{-1})$ | $6.62 \times 10^{-3}$ | $6.15 \times 10^{-4}$ |

[a] $X \in \{2.5, 3.0, 3.5, 4.0, 4.5, 5.0, 5.5, 6.0\}$.

Figure 5 displays concentration profiles in the two films when $m = 0$ (corresponding to zero far-field concentration in the air side). The [diol]/[aldehyde] ratios are in excess of $K_A$ and $K_W$ throughout the air and water films. Similarly, Figure 6 displays concentration profiles in the $m \to \infty$ limit (corresponding to zero far-field concentration in the water side). Then, the [diol]/[aldehyde] ratios fall below $K_A$ and $K_W$.

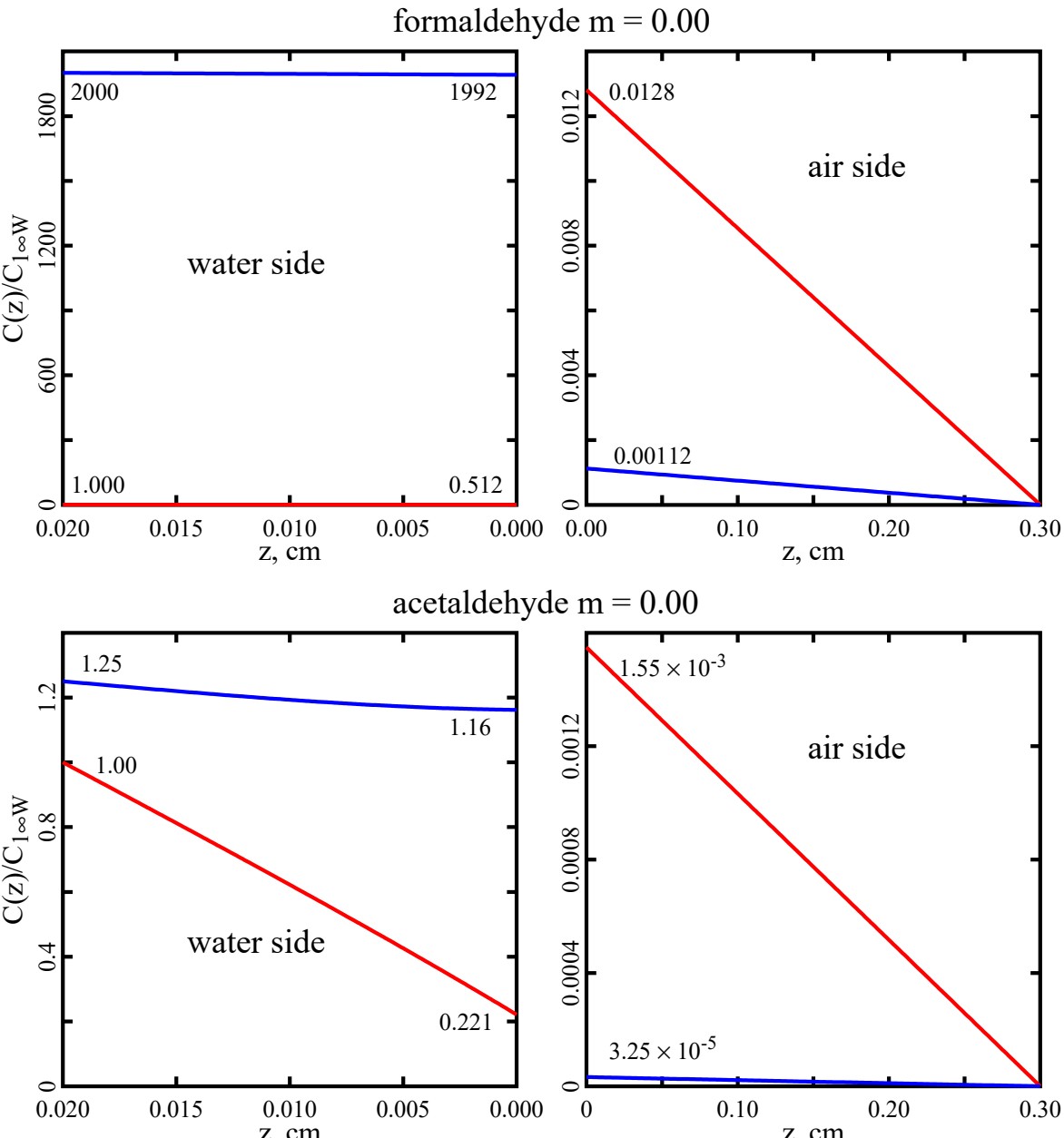

**Figure 5.** Concentration profiles through the films as calculated in model A4 with $m = 0$ and corresponding to the calculations summarized in Table 6. All concentrations are normalized to $C_{1\infty W} = 1$. Red = aldehyde concentration, blue = diol concentration. Intercepts at $z = 0$ and $L$ are shown.

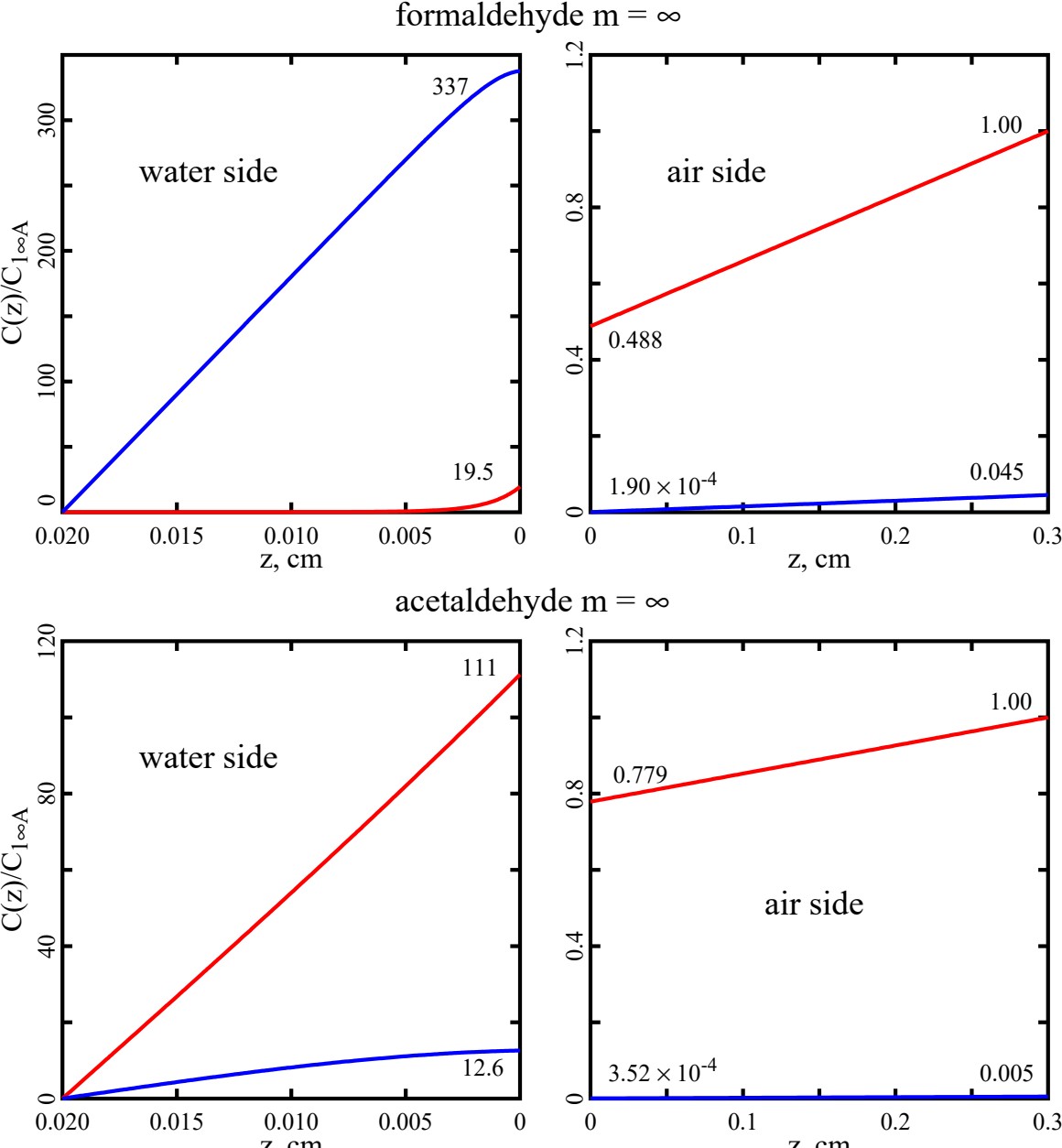

**Figure 6.** Concentration profiles through the films as calculated in model A4 with $m \to \infty$ and corresponding to the calculations summarized in Table 6. All concentrations are normalized to $C_{1\infty A} = 1$. Red = aldehyde concentration, blue = diol concentration. Intercepts at $z = 0$ and $L$ are shown.

### 7.2. Sensitivity to Parameter Variations

Most of the variables in Tables 4 and 5 are sensitive to temperature, humidity, pH, salinity, etc., and are also subject to experimental uncertainty. To judge the impact of such variations on the results, I performed Monte Carlo calculations in which values of the input variables $H_1$, $D_{1W}$, $D_{2W}$, $D_{1A}$, $D_{2A}$, $k_{12W}$, $k_{21W}$, $K_A$, $L_W$, and $L_A$ were sampled independently from log-normal distributions

$$P(X) = \left(2\pi\sigma^2\right)^{-1/2} \exp\left[\frac{-(x-\mu)^2}{2\sigma^2}\right], \quad x = \log_{10} X$$

Consistent with the range of entries in Tables 4 and 5 and defined in Table 7. The mode of a log-normal distribution is at $X = 10^\mu$, while $\sigma$ controls the breadth. $k_{21A}$ was allowed to vary subject to the constraints $k_{21A} < k_{21W}$ and $k_{12A} < k_{12W}$. The variables $K_W$, $k_{12A}$, and $H_2$ were then calculated to enforce constraints given in Table 2. When $\sigma = 0.15$, 95% of the log-normal distribution lies within the range $0.5 \times 10^\mu$ and $2 \times 10^\mu$. Because $K_A$ for formaldehyde is not well known, its $\sigma$ has been set at 0.5, which places 95% of the distribution between one order of magnitude below and one order of magnitude above $10^\mu$. Other less well-constrained variables have been assigned $\sigma$'s of 0.2 or 0.3.

**Table 7.** Distribution of variables used with the formaldehyde and acetaldehyde models.

| Variable | Formaldehyde | | Acetaldehyde | |
|:---:|:---:|:---:|:---:|:---:|
| | $\mu$ | $\sigma$ | $\mu$ | $\sigma$ |
| $H_1$ | −1.6 | 0.15 | −2.2 | 0.15 |
| $D_{1W}/\left(\text{cm}^2\,\text{s}^{-1}\right)$ | −4.7 | 0.15 | −4.8 | 0.15 |
| $D_{2W}/\left(\text{cm}^2\,\text{s}^{-1}\right)$ | −4.8 | 0.15 | −4.9 | 0.15 |
| $D_{1A}/\left(\text{cm}^2\,\text{s}^{-1}\right)$ | −0.8 | 0.15 | −0.9 | 0.15 |
| $D_{2A}/\left(\text{cm}^2\,\text{s}^{-1}\right)$ | −0.9 | 0.15 | −1.0 | 0.15 |
| $k_{12W}/\left(\text{s}^{-1}\right)$ | 1.0 | 0.2 | −2.0 | 0.3 |
| $k_{21W}/\left(\text{s}^{-1}\right)$ | −2.3 | 0.2 | −2.1 | 0.3 |
| $K_A$ | −1.3 | 0.15 | −2.3 | 0.5 |
| $L_W/\text{cm}$ | −1.7 | 0.15 | −1.7 | 0.15 |
| $L_A/\text{cm}$ | −0.5 | 0.15 | −0.5 | 0.15 |
| $k_{21A}$ | Unconstrained, except to enforce $k_{21A} < k_{21W}$ and $k_{12A} < k_{12W}$ | | | |
| $K_W$ | $= k_{12W}/k_{21W}$ | | | |
| $k_{12A}$ | $= K_A\,k_{21A}$ | | | |
| $H_2$ | $= H_1 K_A/K_W$ | | | |

Figure 7 shows the distributions of $f_{A1E}$, $f_{A2}$, $f_{A3}$, and $f_{A4}$ for both the formaldehyde and acetaldehyde models. The results obviously satisfy the inequalities $f_{A2} < f_{A3} < f_{A4} < f_{A1E}$, although the $f_{A2}$, $f_{A3}$ and $f_{A4}$ curves are almost completely superimposable. Figure 8 shows the distributions of $R_A$ for both models. These rather broad distributions, with $R_A$ occurring over much of the available range $0 < R_A < 1$, indicate that neither model has exclusively air- or water-barrier control. Both models are near a tipping point: modulations of less than an order of magnitude in the model parameters significantly shift the balance between air- and water-control. Figure 9 shows the distributions of the ratios $f_{A1E}/f_{A4}$, which confirms the inequality $f_{A4} < f_{A1E}$. $f_{A1E}$ is on the order of 20% and 50% higher than $f_{A4}$ for the two respective models. Figure 10 displays the distribution of $f_{A3}/f_{A4}$ and $f_{A2}/f_{A4}$ for the formaldehyde and acetaldehyde models. The $f_{A3}$ and $f_{A2}$ curves are indistinguishable because $f_{A2}$ is very near $f_{A3}$ for these models. For acetaldehyde, $f_{A2}$ and $f_{A3}$ are typically 94.5% of $f_{A4}$, but distributed broadly, while for formaldehyde, the ratio is about 99.7%.

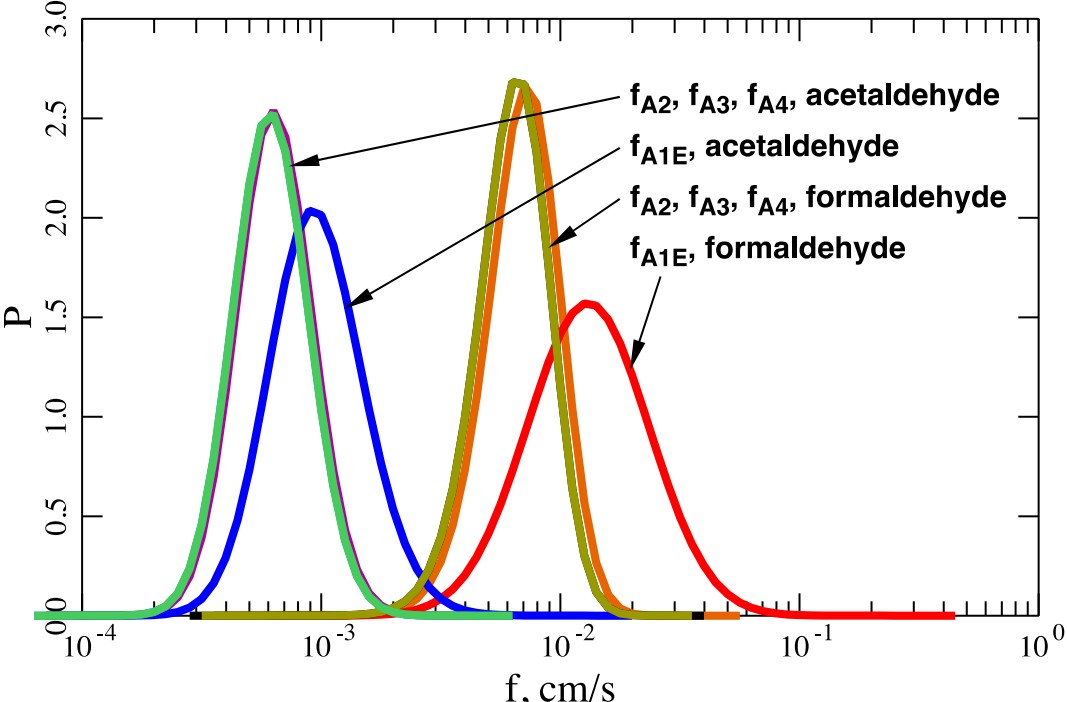

**Figure 7.** Distributions of $f_{A1E}$, $f_{A2}$, $f_{A3}$, and $f_{A4}$ when model parameters are selected according to Table 7. The curves for $f_{A2}$, $f_{A3}$, and $f_{A4}$ are practically identical at the resolution of the figure. Results for $k_{21A}$ between $10^{-6}\text{s}^{-1}$ and $10^{-2.5}\text{s}^{-1}$ have all been included, but also cannot be resolved.

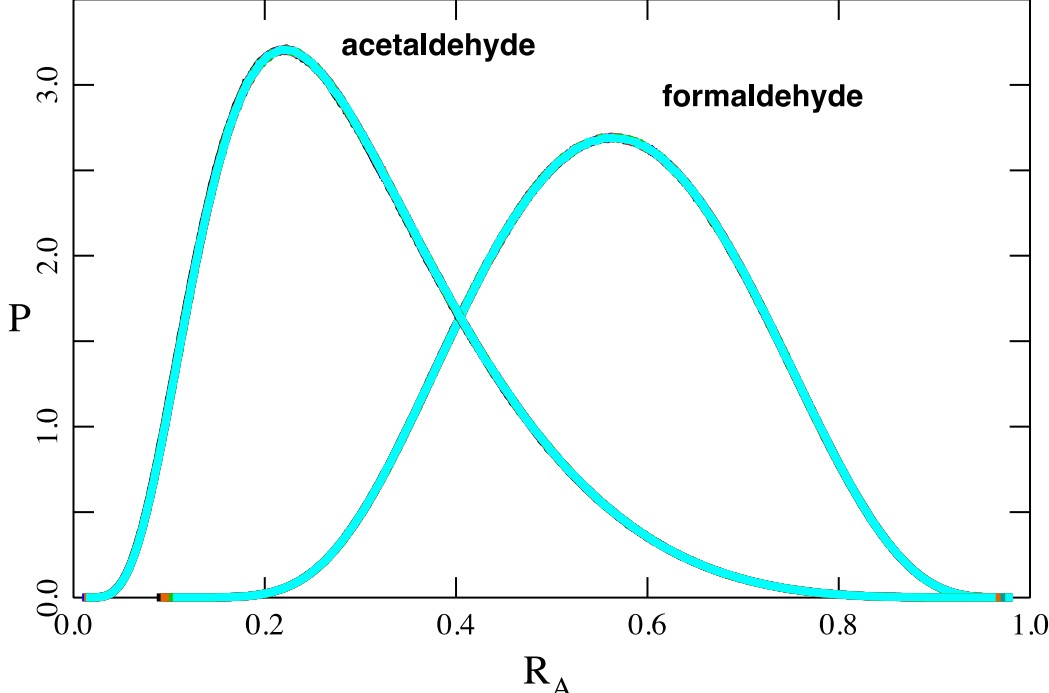

**Figure 8.** Distributions of $R_A$ when model parameters are selected according to Table 7. Individual curves for $k_{21A}$ between $10^{-6}\text{s}^{-1}$ and $10^{-2.5}\text{s}^{-1}$ have all been drawn but lie on top of each other at the resolution of the figure.

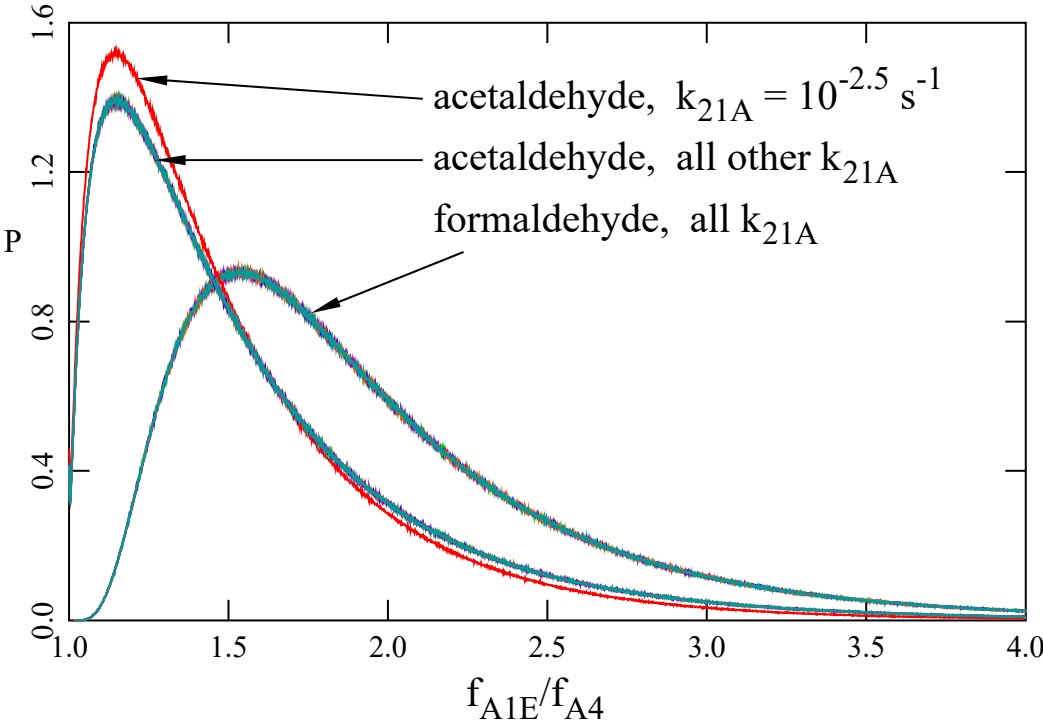

**Figure 9.** Distribution of the ratio $f_{A1E}/f_{A4}$ for the formaldehyde and acetaldehyde models. Curves for $k_{21A}$ between $10^{-6}$ s$^{-1}$ and $10^{-2.5}$ s$^{-1}$ are all displayed, but except for acetaldehyde at $k_{21A} = 10^{-2.5}$ s$^{-1}$, they are practically indistinguishable; 1.8% and 0.6% of the probability density, for formaldehyde and acetaldehyde respectively, are in the tails at $f_{A1E}/f_{A4} > 4$.

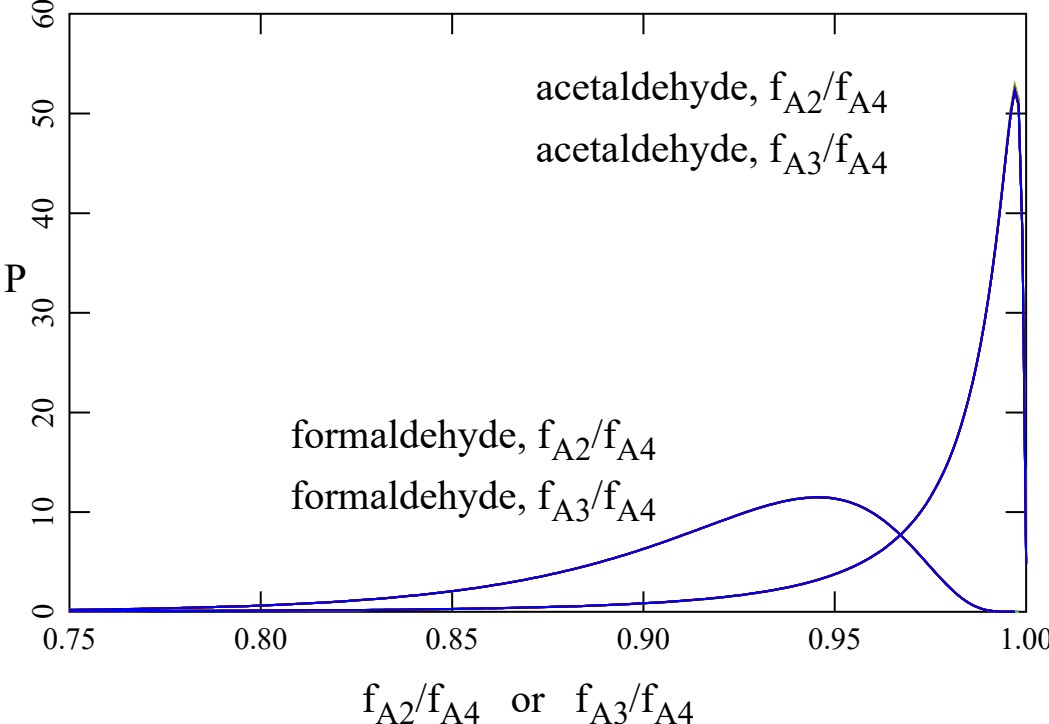

**Figure 10.** Distributions of $f_{A3}/f_{A4}$ and $f_{A2}/f_{A4}$ for the formaldehyde and acetaldehyde models. Curves for eight different values of $k_{21A}$ have been plotted but are indistinguishable, as are the curves for $f_{A2}$ and $f_{A3}$.

*7.3. Comparison with Experiment*

Two separate measurement sets are considered here. (1) Seyfioglu and Odabasi [84] measured transport of formaldehyde from air to water, which corresponds to the $m \to \infty$ limit. In their notation, $F = -K_g C_g$ (with the sign flipped to be consistent with my sign convention). $C_g$ is the gas-phase concentration, equivalent to $C_{1\infty A} + C_{2\infty A}$. In the limit $m \to \infty$ of model A4, $F = -C_{1\infty A} f_{A4}/H_1$. Therefore, the connection between the two notations is $f_{A4} = H_1(1 + K_A)K_g$. (2) Liu et al. [85] measured $H_e$ for formaldehyde at 23 °C to be in the range (0.896 to 1.23) $\times 10^{-5}$, near the consensus value, $1.3 \times 10^{-5}$, reported by Sander [67]. They also measured water-to-air fluxes of formaldehyde. In their notation, $F = K_{0L} C_L$, with $C_L = C_{1\infty W} + C_{2\infty W}$. Compare this with the notation $F = C_{1\infty W} f_{A4}$ in the $m \to 0$ limit of model A4. The connection between the two notations is $f_{A4} = (1 + K_W)K_{0L}$.

Table 8 compares predictions from data in Table 6 against the two sets of measurements. Wind speeds as cited in each paper have been adjusted to $U_{10}$, the wind speed at 10 m above the water surface, using the 1/7-th power law [86]. As expected, the data display variations due to variable wind speed or agitation of the water phase. However, the model predictions are all consistent with the experiment, especially at lower wind speeds. In Section 4, we saw that $L_A$ or $L_W$ are effective film thicknesses, free to be chosen to give agreement with experiment. Therefore, the agreement displayed in Table 8 is mainly a testament to the ability of Schwarzenbach et al. [43] to estimate good $L_A$ and $L_W$ values.

**Table 8.** Comparison with experiment.

| Seyfioglu and Odabasi [84]; Formaldehyde; Air $\to$ Water Experiments; Model A4. | | | |
|---|---|---|---|
| Experimental conditions | $U_{10}/(\mathrm{m\ s^{-1}})$ | $K_g H_1(1 + K_A)/(\mathrm{cm\ s^{-1}})$ | $f_{A4}/(\mathrm{cm\ s^{-1}})$ |
| Laboratory | 8 | $(15.1 \pm 5.5) \times 10^{-3}$ | $(5 \text{ to } 10) \times 10^{-3}$ |
| Field | 4 | $(6.3 \pm 3.1) \times 10^{-3}$ | |
| Liu et al. [85]; Formaldehyde; Water $\to$ Air Experiments; Model A4. | | | |
| Experimental conditions | $U_{10}/(\mathrm{m\ s^{-1}})$ | $K_{0L}(1 + K_W)/(\mathrm{cm\ s^{-1}})$ | $f_{A4}/(\mathrm{cm\ s^{-1}})$ |
| Without stirring | 0.4 | $4.66 \times 10^{-3}$ | |
| | | $4.51 \times 10^{-3}$ | |
| With stirring | 0.4 | $5.61 \times 10^{-3}$ | $(5 \text{ to } 10) \times 10^{-3}$ |
| | | $6.06 \times 10^{-3}$ | |
| | | $6.06 \times 10^{-3}$ | |

## 8. Summary

Volatile gases such as formaldehyde and acetaldehyde undergo hydration reactions in aqueous solution, forming methanediol and 1,1 ethanediol, respectively. Conventional wisdom has dictated that the hydrated forms do not exist in the vapor phase. Therefore, models designed to treat the transport of these compounds between an aqueous phase (oceans, lakes, wastewater lagoons, aqueous aerosols, etc.) and the atmosphere have always assumed that molecules traverse the interface only in the anhydrous form [37,43,53–55]. We now know that the conventional wisdom is wrong—the diols exit in the vapor phase [19–24]. The hydrated molecules are inherently very stable, but catalysts are readily available in the atmosphere leading to chemical equilibrium between both forms. These equilibria strongly favor the anhydrous form [21–23,61,62].

This paper introduces a model of the transport of molecules across the air–water interface that exist in two interconvertible forms, $1 \leftrightarrow 2$, in both phases and in which both forms are assumed to cross the interface. For formaldehyde and acetaldehyde, the former model (A3) predicts fluxes a few percent less that the new one (A4). However, there are conditions for which the differences between the two models may be much greater.

It is also important to emphasize the difference between intrinsic and effective Henry's constants. For formaldehyde and acetaldehyde, the model employing an effective Henry's constant (A1E) overpredicts fluxes by about 90% and 60%, respectively. This is significant because most literature citations report the effective Henry's constant [67].

**Supplementary Materials:** The following are available online at http://www.mdpi.com/2073-4433/11/10/1057/s1, Reacting species models.xlsm. Codes the calculation of flux in models A1E, A2, A3, and A4. When prompted, the user should select "Enable Macros". Reacting species paper supplemental.pdf. Gives derivations and numerical analysis of all models.

**Funding:** This research was funded by [Uintah County Impact Mitigation Special Services District].

**Conflicts of Interest:** The authors declare no conflict of interest.

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
