# Peer review of "Mass Transport of Gases across the Air–Water Interface: Implications for Aldehyde Emissions in the Uinta Basin, Utah, USA"

_atmosphere, doi:10.3390/atmos11101057_

Round 1

Reviewer 1 Report

This is a very useful work on the mass transport of gases across the air-water interface which has been a missing link in understanding the tropospheric fate of diol compounds. The paper is well written and clearly explained. The only question that I have for author is that does this model also consider dimeric forms of compounds? For example, diols can also exist as dimers and seem feasible based on previous calculations. Has author looked at that possibility?

Author Response

There are two approaches that might be considered.  (1)  One could use the current formalism and lump the dimers, etc., in with compound 2.  (2)  One could extend the formalism to include compounds 3, 4, ..., etc. explicitly.  My approach to the issue is summarized on lines 244-246:  At the total aldehyde concentrations expected in the environment, the dimers, etc., are expected to have very small concentrations.  This might become an issue if one wanted to examine laboratory emissions of concentrated formalin solutions, for example, but such elaborations are not necessary for the problem at hand.  Since I already addressed this issue in the manuscript, I have not made any revisions in response to this reviewer.

Reviewer 2 Report

Normally, articles with one author need big improvement because they do not undergo the process of “internal review” between co-authors. However, this paper is surprisingly clear and well written. The author explains in a clear way the problem, the models available, the variables and the choice of the model. Then he goes through an accurate description, the application and the results.

Usually, model description is, literally, boring. But not here! The article is written as a novel and I can only congratulate the author for this.

I have only few suggestions

Line 28: Is this valid also for droplets in equilibrium with the gas phase? Like cloud and fog? Can you add some reference?

Lines 35-39: In chemistry, the bifrontal arrow is used to indicate resonance structures, while the double arrow to indicate the equilibrium. I think that in this paper all the bifrontal arrows should be changed into double arrows.

Lines 46-47: could you provide a reference?

Line 51: what do you mean with “intact”?

Lines 284-285: some recent experimental works report a huge deviation from the Henry’s equilibrium for less polar compounds also with carbonyl functions (Van Pinterxen 2005, Wang 2019, Atmos Res), showing that they are more soluble than expected. What are the results for the simulations with the highest values of KH?

Line 548: there is a typo in the beginning of the reference

Supplementary material: please number each section and make references to the specific paragraphs.  

Author Response

First suggestion:  I added the phrase "aqueous aerosols" at line 399 of the Summary.

Second suggestion:  All those arrows have been changed to double arrows.

Third suggestion:  References have been added following the sentence "Many measurements have detected ..."

Fourth suggestion:  I have changed the phrase "as intact molecules" to "in the diol form."

Fifth suggestion:  As in the original Whitman model, the major impact of tuning Henry's constant is to switch between air- and water-barrier control.  This is addressed in the manuscript at line 360 where I characterize the behavior as a tipping point.

Sixth suggestion:  Are you referring to SF6 and 3He in the Clark et al. title?  Those typos have been fixed.